# PLGC: Pseudo-Labeled Graph Condensation

**Jay Nandy**[*]                                                                              *jayjaynandy@gmail.com*
*Fujitsu Research of India, Bangalore*

**Arnab Kumar Mondal**[†]                                                    *arnabkumarmondal123@gmail.com*
*Fujitsu Research of India, Bangalore*

**Anuj Rathore**                                                                            *anuj.rathore@fujitsu.com*
*Fujitsu Research of India, Bangalore*

**Mahesh Chandran**                                                              *mahesh.chandran@fujitsu.com*
*Fujitsu Research of India, Bangalore*

**Reviewed on OpenReview:** *https://openreview.net/forum?id=TkpewrzsnJ*

## Abstract

Large graph datasets make training graph neural networks (GNNs) computationally costly. Graph condensation methods address this by generating small synthetic graphs that approximate the original data. However, existing approaches rely on clean, supervised labels, which limits their reliability when labels are scarce, noisy, or inconsistent. We propose Pseudo-Labeled Graph Condensation (PLGC), a self-supervised framework that constructs latent pseudo-labels from node embeddings and optimizes condensed graphs to match the original graph's structural and feature statistics-without requiring ground-truth labels. PLGC offers three key contributions: (1) A diagnosis of why supervised condensation fails under label noise and distribution shift. (2) A label-free condensation method that jointly learns latent prototypes and node assignments. (3) Theoretical guarantees showing that pseudo-labels preserve latent structural statistics of the original graph and ensure accurate embedding alignment. Empirically, across node classification and link prediction tasks, PLGC achieves competitive performance with state-of-the-art supervised condensation methods on clean datasets and exhibits substantial robustness under label noise, often outperforming all baselines by a significant margin. Our findings highlight the practical and theoretical advantages of self-supervised graph condensation in noisy or weakly-labeled environments. [1].

## 1 Introduction

Large-scale graph-structured data arise in numerous domains, including social networks (Fan et al., 2019), biological systems (Muzio et al., 2021), and knowledge graphs (Baek et al., 2020). The growing scale and complexity of these graphs, often comprising millions of nodes with high-dimensional attributes and heterogeneous connectivity patterns, pose significant computational challenges for training graph neural networks (GNNs). These challenges are exacerbated in resource-constrained environments and in settings requiring repeated training, such as hyperparameter search, architecture exploration, or continual adaptation to dynamic graph streams.

Graph condensation has emerged as an effective strategy to mitigate these limitations by synthesizing a compact graph whose structural and feature distributions approximate those of the original data (Jin et al., 2022b; Zheng et al., 2023; Zhang et al., 2024; Yang et al., 2024; Sun et al., 2024). Models trained on the condensed graph attain accuracy close to those trained on the full graph while reducing training and storage

---

[*]Current affiliation: eBay India; [†]Current affiliation: IBM India
[1]Code Link: https://github.com/jayjaynandy/PLGC

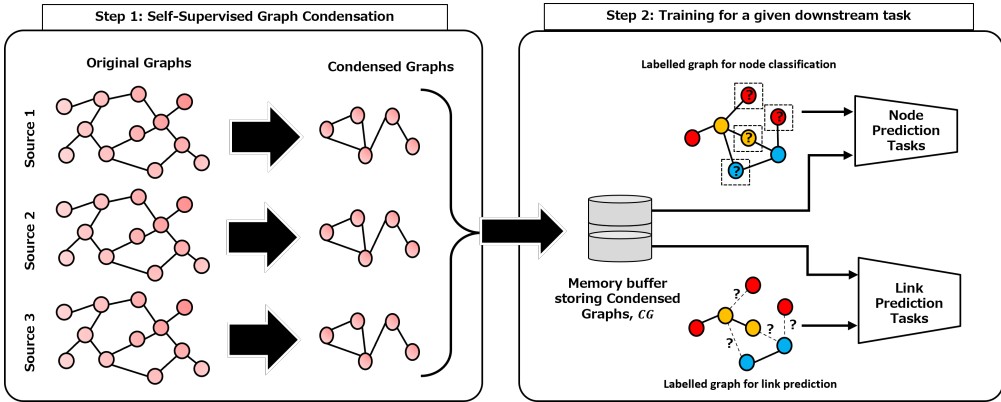

Figure 1: Proposed self-supervised graph condensation framework to condense the original (unlabeled) graphs from different distributed sources within a small memory that can be used for training or finetuning for downstream tasks with limited supervision.

costs substantially. Despite these advantages, existing graph condensation methods predominantly depend on high-quality supervised labels to guide the construction of the synthetic graph. This reliance significantly restricts their applicability. In many real-world scenarios—financial transaction networks (Sheng et al., 2011), dynamic social interactions (Wang et al., 2015), and sensor or IoT graphs (Yick et al., 2008; Dai et al., 2021; Ding et al., 2024)—labels are scarce, incomplete, noisy, or subject to temporal distribution shift. Such conditions undermine the stability and effectiveness of label-dependent condensation objectives.

To address these limitations, we propose **Pseudo-Labeled Graph Condensation (PLGC)**, a self-supervised condensation framework that circumvents the need for ground-truth labels by constructing and optimizing *latent pseudo-labels* that capture the intrinsic structural and feature characteristics of the original graph. PLGC operates through an alternating optimization procedure consisting of two components:

1. **Pseudo-Label Construction**, in which latent soft labels are inferred by encoding statistical relationships in node features and neighborhood structures; and

2. **Condensed Graph Optimization**, where the synthetic graph is updated by minimizing the divergence between its embedding distribution and the pseudo-label-induced representation of the original graph.

This alternating mechanism ensures that the synthetic graph remains aligned with the latent geometry of the original graph, regardless of supervised information. Furthermore, PLGC extends naturally to **multi-source condensation**, where multiple independent graphs are available. In such cases, PLGC learns source-specific condensed graphs while jointly aligning their latent representations through pseudo-labels, thereby enabling the training of a unified downstream model in a fully or partially unsupervised manner (Figure 1). Unlike prior graph condensation methods, PLGC introduces a tightly coupled, closed-loop framework that jointly learns pseudo-labels and condensed graph representations with explicit alignment, enabling fully label-free condensation without sacrificing downstream transferability.

A key contribution of this work is a **theoretical characterization of pseudo-label stability and condensation error propagation**. We establish (i) conditions under which pseudo-label estimation remains stable in the presence of noise or distributional variation, and (ii) upper bounds on the embedding divergence between the original and condensed graphs induced by PLGC's optimization dynamics. These results formally justify the robustness of PLGC in scenarios where supervised condensation methods deteriorate.

Our primary contributions are as follows:

1. We identify structural limitations of supervised graph condensation, demonstrating analytically and empirically that their effectiveness deteriorates under label noise, label scarcity, and label distribution shift.

2. We introduce **PLGC**, a self-supervised condensation framework based on alternating pseudo-label estimation and condensed-graph optimization, completely eliminating the need for ground-truth labels.

3. We provide **theoretical guarantees**, including stability results for pseudo-label inference and error bounds for condensed–original embedding alignment, establishing principled robustness under weak or unreliable supervision.

4. We conduct extensive experiments on five benchmark datasets, showing that (i) PLGC significantly outperforms supervised condensation methods in low-label, noisy-label, and shift-label regimes; (ii) it remains competitive with supervised methods when clean labels are available; and (iii) it offers superior generalization in multi-source and distributionally heterogeneous settings.

## 2 Background & Related Work

### 2.1 Problem Formulation

We consider a setting with $M$ independent sources, each providing an unlabeled graph $\{\mathcal{T}^i\}_{i=1}^M$ defined over a shared *node-level* label space. Each graph $\mathcal{T}^i = (X^i, A^i)$ contains $N^i$ nodes with node features $X^i \in \mathbb{R}^{N^i \times d}$ and adjacency matrix $A^i \in \{0,1\}^{N^i \times N^i}$, but no ground-truth labels. The objective is to construct, for each source graph, a significantly smaller *condensed graph* paired with a set of *pseudo-labels*, $\{(\mathcal{S}^i, \widetilde{Y}^i)\}_{i=1}^M$, such that the condensed graphs retain the essential structural and feature-level information required for downstream predictive tasks.

Formally, each condensed graph $\mathcal{S}^i = (\widetilde{X}^i, \widetilde{A}^i)$ contains $N'^i$ synthetic nodes, where $N'^i \ll N^i$, and is associated with pseudo-labels $\widetilde{Y}^i = \{\widetilde{y}_k^i\}_{k=1}^{N'^i}$ that summarize the latent cluster-level statistics of the original graph. Our goal is to learn a representation model $\text{GNN}_\theta$ such that:

1. **Condensed graph alignment.** The condensed graphs can be used to train $\text{GNN}_\theta$ in a self-supervised manner by aligning node embeddings with the pseudo-labels:

$$\text{GNN}_\theta(\mathcal{S}^i) \to \widetilde{Y}^i, \quad \forall\, i \in \{1, \ldots, M\}.$$

2. **Downstream transferability.** Once $\text{GNN}_\theta$ is learned from synthetic graphs, it can be transferred to a new downstream task by fine-tuning only a lightweight prediction head $f_\phi$ using limited supervision:

$$f_\phi\big(\text{GNN}_\theta(\mathcal{G}^{\text{test}})\big) \to Y^{\text{test}},$$

where $\mathcal{G}^{\text{test}}$ is a test graph with available labels.

Thus, the overall goal is to design a self-supervised graph condensation framework that produces compact synthetic graphs and pseudo-labels that preserve task-relevant latent structure, enabling efficient and minimally supervised training of downstream graph models.

***Graph reduction techniques*** aim to reduce the size of a given graph while preserving essential information, and can be broadly categorized into graph sparsification, coarsening, sketching, and graph condensation. Graph sparsification selects a subset of nodes or edges from the original graph to reduce complexity (Chen et al., 2021; Razin et al., 2023). Graph coarsening constructs a smaller graph by learning a surjective mapping from the original graph, typically by merging multiple nodes into supernodes (Si et al., 2022; Kumar et al., 2023). Graph sketching summarizes the original graph into a compact representation by preserving selected structural properties (*e.g.*, centrality or spectral statistics) through node or edge sampling (Ding et al., 2022).

Although these approaches are often unsupervised or require minimal supervision, they face notable limitations in practical learning scenarios. For instance, graph sparsification becomes less effective when nodes carry rich attributes, and coarsening or sketching methods typically preserve task-agnostic properties (*e.g.*, eigenvalues or degree distributions) that may not align with downstream objectives. As a result, their performance often lags behind graph condensation methods when used for training graph neural networks.

| | Objective | Condensed Memory | Self supervision | Multiple Sources | Downstream Performance | Robustness under label-noise |
|---|---|---|---|---|---|---|
| **Supervised Condensation (Existing)** | Produce a synthetic graph by preserving class-specific characteristics. | Yes | No | Yes | Good | Poor |
| **Coarsening & Sparsification** | Returns a small graph by preserving certain graph properties | Yes | Yes | Yes | Poor | Poor |
| **Graph SSL** | Learns a generalized representation model for various downstream tasks. | No | Yes | No | Good | Good |
| **PLGC (Ours)** | Returns a small synthetic graph by preserving latent statistics. | Yes | Yes | Yes | Good | Good |

Table 1: Advantages of our pseudo-labeled-based self-supervised graph condensation over existing techniques.

***Graph condensation (supervised)*** methods aim to distill large graphs into small synthetic datasets that preserve latent structural and semantic information, such that models trained on condensed graphs and perform comparably to those trained on the full data (Jin et al., 2022b;a; Liu et al., 2022; Zheng et al., 2023; Zhang et al., 2024; Liu et al., 2023a;b; Lei & Tao, 2023; Yu et al., 2023). Early work focused on gradient matching between real and synthetic data (Wang et al., 2018; Nguyen et al., 2020; Zhao et al., 2020), initially in continuous domains (e.g., images) and later extended to graphs.

GCond (Jin et al., 2022b) adapts online gradient-matching for graph-structures by jointly learning node features and adjacency matrices. DosCond (Jin et al., 2022a) improves efficiency via single-step gradient matching using probabilistic graph models for graph classification. Subsequent approaches further reduce computation by implicitly encoding topology into node features, often using identity or fixed adjacency structures. Representation matching methods, *e.g.,* CaT (Liu et al., 2023a), PUMA (Liu et al., 2023b), SERGCL (Mondal et al., 2024) generate condensed graphs by aligning class-wise representations between real and synthetic graphs. Trajectory-based approaches, including SFGC (Zheng et al., 2023) and GEOM (Zhang et al., 2024), leverage expert training trajectories from original graphs to guide condensation. Although effective, these methods incur substantial computational and memory overhead due to the storage and replay of their expert trajectories (Appendix B). ST-GCOND (Yang et al., 2025) improves efficiency through spectral approximations, but remains fundamentally supervised, limiting its robustness in noisy or low-label settings.

A fundamental limitation of most of these existing graph condensation methods is their reliance on labeled data. In practice, node or graph labels can be noisy, incomplete, or unavailable, significantly degrading performance. Even automatically generated labels can be corrupted by measurement errors due to faulty sensors or distribution shifts in evolving networks (Carrera & Kim, 2020; Nandy et al., 2020; Chauhan et al., 2022; Bazhenov et al., 2024).

***Graph Condensation & Self-Supervised Learning (SSL).*** Graph SSL enables representation learning without labeled supervision and has demonstrated strong generalization across downstream tasks (Kim & Oh, 2021; You et al., 2020; Nandy et al., 2024). SSL methods generally fall into two categories: (1) *predictive approaches* (Hu et al., 2020b; Kim & Oh, 2021; Rong et al., 2020), which construct pretext tasks based on structural signals and emphasize local semantics, and (2) *contrastive learning methods* (You et al., 2020; 2021; Yin et al., 2022), which maximize agreement between augmented views to capture global semantics. However, GSSL methods typically produce a backbone GNN model rather than condensed datasets, therefore, do not compress data or aggregate multi-source knowledge.

Recent work has begun exploring self-supervised formulations for graph condensation, although important gaps remain. SGDC (Wang et al., 2024) performs condensation for graph-level datasets by matching representations from a fixed self-supervised encoder. While effective for graph classification, it does not model node-level correspondence between the latent structure and the condensed data and, therefore, does not directly support tasks requiring node-level pseudo-label alignment or downstream fine-tuning with limited supervision. CTGC (Gao et al., 2025) incorporates contrastive objectives to disentangle semantic and structural signals, yet it does not construct explicit prototype-based pseudo-labels nor provide theoretical guarantees on cluster preservation or assignment stability.

In contrast, PLGC instantiates this framework through a principled pseudo-labeling mechanism that jointly learns latent prototypes and node–prototype assignment matrices in a fully self-supervised manner. This formulation enables direct alignment between condensed nodes and pseudo-labels, supports stable representation matching with theoretical guarantees on cluster separation and centroid concentration, and yields strong robustness under label noise. Furthermore, PLGC extends naturally to multi-source settings by aligning source-specific condensed graphs within a shared latent space, enabling transferable representations across heterogeneous data sources.

While PLGC builds upon concepts from self-supervised learning and clustering, we emphasize that its contribution is not a straightforward extension of these ideas but rather their principled integration within the graph condensation setting. Existing graph condensation methods are predominantly supervised and rely explicitly on ground-truth labels during the condensation process. In contrast, prior self-supervised condensation approaches, such as SGDC and CTGC, either do not construct explicit pseudo-labels tied to condensed nodes or do not provide a unified optimization framework that jointly learns pseudo-labels and condensation variables. PLGC addresses these limitations through a closed-loop formulation in which (i) pseudo-labels are learned directly from the original graph, (ii) each condensed node is explicitly aligned with a unique pseudo-label via a one-to-one mapping, and (iii) pseudo-label learning and condensed graph optimization are coupled through an alternating refinement procedure. This tight coupling is critical for enabling fully label-free condensation while preserving downstream transferability, an aspect that prior work does not address.

Table 1 summarizes the strengths and limitations of existing methods. While graph condensation has broad applications—including architecture search, privacy preservation, adversarial robustness, and continual learning—prior approaches either rely heavily on labels or lack reusable condensed representations. PLGC bridges this gap by combining the generalization strength of SSL with the efficiency of condensation, enabling reusable condensed graphs and effective knowledge aggregation from noisy or unlabeled multi-source data.

## 3 Proposed Method

Existing graph condensation methods are typically supervised, that rely on ground-truth labels to guide either gradient matching or representation matching during the construction of a synthetic graph (Jin et al., 2022b; Liu et al., 2023a). Their objective can be written as

$$\min_{\mathcal{S}} \ \mathcal{L}_{node}(\text{GNN}_{\theta_{\mathcal{S}}}(A, X), Y) \quad \text{s.t.} \quad \theta_{\mathcal{S}} = \arg\min_{\theta} \ \mathcal{L}_{node}(\text{GNN}_{\theta}(A', X'), Y'), \tag{1}$$

where $\mathcal{T} = (A, X, Y)$ denotes the original labeled graph and $\mathcal{S} = (A', X', Y')$ is the condensed graph with $N' \ll N$ nodes. Here, $A \in \mathbb{R}^{N \times N}$ and $A' \in \mathbb{R}^{N' \times N'}$ denote adjacency matrices; $X \in \mathbb{R}^{N \times d}$ and $X' \in \mathbb{R}^{N' \times d}$ denote node features; $\text{GNN}_{\theta}$ is a GNN parameterized by $\theta$; and $\mathcal{L}_{node}$ is a supervised node-classification loss (e.g., cross-entropy).

The goal in Eq. equation 1 is to learn a synthetic graph $\mathcal{S}$ such that the model trained on $\mathcal{S}$ generalizes to the original graph $\mathcal{T}$. However, the dependence on clean labels $Y$ renders these methods fragile in noisy-label scenarios (Chauhan et al., 2022). In contrast, designing a *self-supervised* condensation objective requires eliminating all dependencies on $Y$, still generating a condensed graph that preserves task-relevant structure.

We address this challenge by learning (i) a set of latent *pseudo-labels* $\tilde{Y}$ and (ii) an associated *assignment matrix* $Q_{\mathcal{T}}$, enabling self-supervised condensation without ground-truth labels.

### 3.1 Pseudo-Labeled Graph Condensation

Our method incorporates two key iterative steps for our self-supervised graph condensation:

1. **Updating $K$ pseudo-labels** $\tilde{Y} \in \mathbb{R}^{K \times d}$, where $K \ll N$, so that each pseudo-label captures representative statistics of one latent cluster in the embedding space.

2. **Updating assignment matrix** $Q_{\mathcal{T}} \in \{0, 1\}^{N \times K}$, where each row is a one-hot vector linking each node to exactly one pseudo-label.

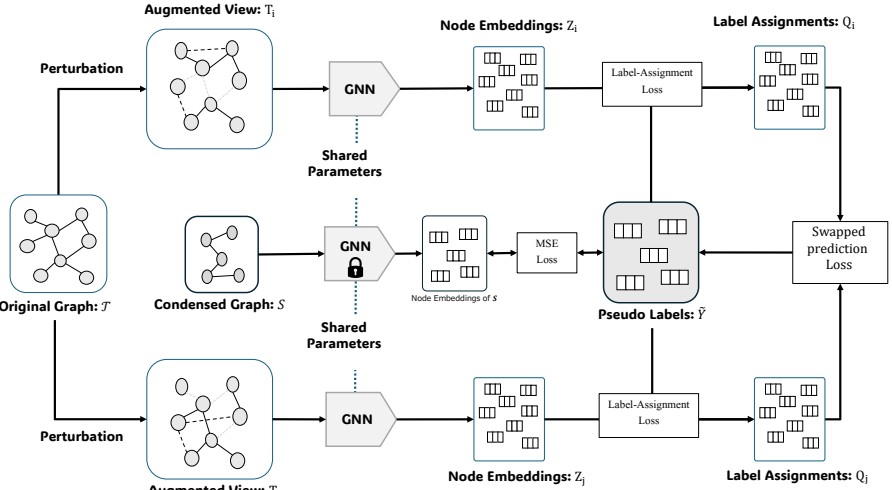

Figure 2: Overview of the proposed PLGC framework. The method alternates between two coupled optimization steps: **(I) Pseudo-Label Learning** — multiple graph augmentations are generated from the original graph, processed through a shared GNN encoder to obtain node embeddings, and assigned to the pseudo-labels using an entropy-regularized Sinkhorn optimization. The resulting assignments and pseudo-labels are updated via a swapped-assignment view prediction loss. **(II) Condensed Graph Optimization** — the condensed graph is passed through the same shared encoder, and its node embeddings are aligned with the learned pseudo-labels using an MSE-based representation matching loss. Together, these steps iteratively refine pseudo-labels and the condensed graph so that the synthetic graph preserves the latent structure of the original graph without relying on ground-truth labels.

The pseudo-label matrix $\tilde{Y} = [\tilde{y}_1, \ldots, \tilde{y}_K]^\top$ represents $K$ distinct pseudo-labels, each capturing a cluster-level embedding in the original graph. Given $Q_\mathcal{T}$, the induced pseudo-label for each node is $Q_\mathcal{T}\tilde{Y}$.

**Self-supervised condensation objective.** We formulate the PLGC objective as an unsupervised analogue of Eq. equation 1, adding two auxiliary optimization problems to learn $Q_\mathcal{T}$ and $\tilde{Y}$:

$$\min_\mathcal{S} \ \mathcal{L}\big(\text{GNN}_{\theta_\mathcal{S}}(A, X), Q_\mathcal{T}\tilde{Y}\big) \quad \text{s.t.} \quad \theta_\mathcal{S} = \arg\min_\theta \ \mathcal{L}\big(\text{GNN}_\theta(A', X'), Q_\mathcal{S}\tilde{Y}\big), \tag{2}$$

where $\mathcal{L}$ measures the discrepancy between node embeddings and pseudo-labels.

$$Q_\mathcal{T}, \tilde{Y} = \arg\min_{Q_\mathcal{T}, \tilde{Y}} \mathcal{L}_{pseudo}\big(\text{GNN}_{\theta'}(A, X), Q_\mathcal{T}\tilde{Y}\big). \tag{3}$$

where $\mathcal{L}_{pseudo}$ is a self-supervised loss used to learn latent clusters.

Because every pseudo-label must correspond to at least one node in $\mathcal{S}$, we set $N' = K$ and fix the condensed-graph assignment matrix as $Q_\mathcal{S} = I_K$, ensuring a one-to-one mapping between nodes of $\mathcal{S}$ and pseudo-labels. Figure 2 illustrates the PLGC pipeline.

### 3.1.1 Learning Pseudo-Labels and Assignment Matrix

We construct $\mathcal{L}_{pseudo}$ using two iterative components: (a) a *swapped-assignment view prediction loss* $\ell_{swap}$, used to update $\tilde{Y}$ and the encoder $\text{GNN}_{\theta'}$, and (b) An *assignment loss* $\ell_{assign}$, used to compute $Q_\mathcal{T}$ by enforcing balanced assignment of nodes to the pseudo-labels. We apply two stochastic augmentations $T_i$ and $T_j$ to the original graph and extract $\ell_2$-normalized embeddings: $\mathcal{Z}_i = \frac{\text{GNN}_{\theta'}(T_i)}{\|\text{GNN}_{\theta'}(T_i)\|_2}$ and $\mathcal{Z}_j = \frac{\text{GNN}_{\theta'}(T_j)}{\|\text{GNN}_{\theta'}(T_j)\|_2}$.

**Swapped-assignment view prediction loss** contrasts between the embeddings of different views by comparing their pseudo-label assignments instead of their features, allowing to update the pseudo-labels in an online fashion (Caron et al., 2020). Given node embedding $z_{i,n}$ from augmentation $T_i$ and pseudo-label

$q_{j,n}$ computed from $T_j$ and a temperature parameter $\tau$, the loss encourages consistent cluster assignments across augmentations:

$$\ell_{swap}(z_{i,n}, q_{j,n}) = -\sum_k q_{j,n}^{(k)} \log \frac{\exp(\frac{1}{\tau} z_{i,n}^\top \tilde{y}_k)}{\sum_{k'} \exp(\frac{1}{\tau} z_{i,n}^\top \tilde{y}_{k'})}. \tag{4}$$

**Balanced assignment loss** ensures that the node-embeddings are equally distributed among the pseudo-labels in a batch are distinct, therefore preventing the trivial solution of producing same assignments for each node. Following (Asano et al., 2019; Caron et al., 2020), we compute $Q_i$ for augmentation $T_i$ by solving

$$\ell_{assign} = \text{Tr}(Q_i^\top \tilde{Y}^\top \mathcal{Z}_i) + \epsilon \, \mathcal{H}(Q_i), \quad \text{s.t.} \quad Q_i \in \left\{ Q_i \geq 0 : Q_i^\top \mathbf{1}_B = \tfrac{1}{K}\mathbf{1}, \quad Q_i \mathbf{1}_K = \tfrac{1}{B}\mathbf{1} \right\}, \tag{5}$$

where $B$ is the batch size and $\mathcal{H}$ is entropy. The solution is given by $Q_i^* = \text{diag}(u)\exp(\tilde{Y}\mathcal{Z}_i/\epsilon)\text{diag}(v)$, with $u, v$ obtained via Sinkhorn-Knopp normalization (Cuturi, 2013). Finally, each row is discretized to the closest one-hot vector. The full pseudo-label learning loss is as follows:

$$\mathcal{L}_{pseudo} = \sum_{i,j} \sum_n \ell_{swap}(z_{i,n}, q_{j,n}) \quad \text{s.t.} \quad Q_i = \arg\max_Q \ell_{assign}. \tag{6}$$

### 3.1.2 Generating Condensed Graphs

After learning $(Q_\mathcal{T}, \tilde{Y})$ and $\text{GNN}_{\theta'}$ using $\mathcal{L}_{pseudo}$, we construct $\mathcal{S}$ by minimizing a representation-matching objective. While prior condensation methods match gradients (Jin et al., 2022b) or class-conditional distributions (Liu et al., 2023a), we adopt latent representation matching via *maximum mean discrepancy (MMD)* (Zhao & Bilen, 2023). The supervised version aligns conditional means, $\mathbb{E}[Z_{\mathcal{T}|y}]$ and $\mathbb{E}[Z_{\mathcal{S}|y}]$ for each $y$.

For PLGC, we replace class labels with pseudo-labels and use $\text{GNN}_{\theta'}$ as the embedding model. Because $Q_\mathcal{S} = I_K$ and $N' = K$, each pseudo-label corresponds to a unique synthetic node. Thus, the MMD objective reduces to a mean-square loss:

$$\min_\mathcal{S} \sum_{\tilde{y} \in \tilde{Y}} \left\| \tilde{y} - z_{\mathcal{S}|\tilde{y}} \right\|^2, \tag{7}$$

where $z_{\mathcal{S}|\tilde{y}}$ denotes the embedding of the synthetic node mapped to $\tilde{y}$. Consistent with findings in (Liu et al., 2023a; Zheng et al., 2023; Zhang et al., 2024), we omit adjacency learning and condense only the node features, benefiting both performance and storage efficiency.

### 3.2 Backbone Reconstruction and Fine-Tuning

Given $M$ sources, PLGC produces $M$ condensed graphs with associated pseudo-labels: $CG = \{(\mathcal{S}_i, \tilde{Y}_i)\}_{i=1}^M$.

**(a) Backbone reconstruction.** We train a backbone GNN by aligning embeddings of condensed graphs with their pseudo-labels: $\min_\theta \sum_{(\mathcal{S}_i, \tilde{Y}_i) \in CG} \sum_{\tilde{y} \in \tilde{Y}_i} \left\| \tilde{y} - z_{\mathcal{S}_i|\tilde{y}} \right\|^2$.

**(b) Task-specific fine-tuning.** For downstream applications, we fine-tune a prediction head $f_\phi$ on task-specific graph $\mathcal{G}^{test}$ using clean labels $Y^{test}$: $\min_\phi \mathcal{L}_{downstream}\big(f_\phi \circ \text{GNN}_\theta(A^{test}, X^{test}), Y^{test}\big)$.

Therefore, PLGC requires no labels during condensation and uses only minimal supervision during downstream fine-tuning.

## 4 Theoretical Motivation and Statement

For graph condensation to remain effective across downstream tasks and varying noise levels, the synthetic graph must faithfully preserve the statistical geometry of the original graph. Since PLGC operates without

access to any domain specific ground-truth class labels, the quality of the learned pseudo-labels is critical – we should ensure the pseudo-labels remain aligned with the underlying latent structure to obtain informative, stable, and robust condensed graphs. Our theoretical analysis formalizes this intuition under the following mild and widely adopted assumptions in centroid-based clustering theory:

A1. **Sub-Gaussian Latent Structure:** A spherical sub-Gaussian latent structure, where node embeddings within each latent cluster exhibit Gaussian-type tail decay and concentration. In particular, sub-Gaussian random variables satisfy exponential tail bounds, exhibiting strong concentration phenomena in high dimensions (Vershynin, 2018; Spokoiny, 2023). It naturally generalizes Gaussian mixture models and encompasses a broader class of distributions, including bounded-support and sub-Gaussian mixtures, which are widely adopted in high-dimensional cluster analysis (Qing, 2026; Bouveyron et al., 2007).

A2. **Separability:** Underlying true latent centers, $\{\mu_k\}$ are well-separated by a non-trivial minimum distance: $\Delta = \min_{j \neq k} \|\mu_j - \mu_k\|$.

Under these conditions, we show that the pseudo-labels $\widetilde{Y} = \{\widetilde{y}_1, \ldots, \widetilde{y}_K\}$ generated by PLGC reliably capture the latent structure of the original graph. In essence, we show that the pseudo-labels remain close to the true latent centers and preserve the separation between pseudo-labels. More formally,

**Theoretical statement and assumptions.** Consider node embeddings $\{z_i\}_{i=1}^n$ sampled independently corresponding to $K$ latent centers, with pseudo-label assignments denoted by $q_{ik} \in \{0, 1\}$. We set the deviation parameter as: $\epsilon_k := 4\sigma\sqrt{\frac{d+\log(2K/\delta)}{s_k}}$ where $s_k := \sum_i q_{ik}$ denotes the effective sample size of the cluster $k$. Assuming (A1) and (A2) and given the pseudo-labels satisfying the stationarity condition (i.e., each pseudo-label aligns with its assignment-weighted feature mean), the following statements hold simultaneously for all $k$ with probability at least $1 - \delta$:

 i. **Pseudo-label Concentration.** Each pseudo-label satisfies $\|\widetilde{y}_k - \mu_k\| \leq \epsilon_k$, i.e., the pseudo-labels produced by PLGC remain tightly concentrated in the true centers of the population.

 ii. **Interior-point recovery.** Any point $z_i$ assigned to $k$-th pseudo-label (i.e., $q_{ik} = 1$) such that $\|z_i - \mu_k\| < \frac{\Delta}{2} - \epsilon_k$ remains closer to $\widetilde{y}_k$ than other labels $\widetilde{y}_\ell$ – indicating that, the interior points are always correctly assigned.

 iii. **Sample Complexity & Pseudo-Label Separation.** A sufficient sample size to ensure a positive interior margin of $\epsilon_{\max} \leq \Delta/\beta$ is: $s_k \geq \frac{16\sigma^2\beta^2}{\Delta^2}\left(d + \log(2K/\delta)\right)$, and leads to a stronger separation of pseudo-labels: $\|\widetilde{y}_k - \widetilde{y}_\ell\| \geq (1 - \frac{2}{\beta})\Delta \quad \forall k \neq \ell$

The complete proofs and additional supporting lemmas are provided in the appendix. These results guarantee that pseudo-labels are well separated, accurately concentrated, and robustly recover interior nodes—ensuring that PLGC preserves the underlying latent structure of the full graph.

## 5 Experiments

**Datasets, Tasks and Evaluation Protocol.** We evaluate PLGC on node classification and link prediction tasks under both transductive and inductive settings. We use five benchmark datasets: Cora, Citeseer

| Datasets | #Nodes | #Edges | #Class | #Feature | Train/Val/Test |
|---|---|---|---|---|---|
| Citeseer | 3,327 | 4,732 | 6 | 3,703 | 120/500/1000 |
| Cora | 2,708 | 5,429 | 7 | 1,433 | 140/500/1000 |
| Ogbn-arxiv | 169,343 | 1,166,243 | 40 | 128 | 90,941/29,799/48,603 |
| Flickr | 89,250 | 899,756 | 7 | 500 | 44,625/22,312/22,313 |
| Reddit | 232,965 | 57,307,946 | 41 | 602 | 15,3932/23,699/55,334 |

Table 2: Statistics of the datasets.

(Kipf & Welling, 2016), and Ogbn-Arxiv (Hu et al., 2020a) for transductive evaluation, and Flickr (Zeng et al., 2019) and Reddit (Hamilton et al., 2017) for inductive evaluation (see Table 2). All experiments follow standard data splits and evaluation protocols for each dataset. Results are averaged over 10 runs, and we report mean performance with standard deviation.

| | r | Supervised | | | | | Self-Supervised | | | | Whole Dataset |
|---|---|---|---|---|---|---|---|---|---|---|---|
| | | DCGraph | GCond | GCond-X | SFGC | GEOM | Random | Herding | K-Center | PLGC (Ours) | |
| Citeseer | 0.9% | $66.8_{\pm1.5}$ | $70.5_{\pm1.2}$ | $71.4_{\pm0.8}$ | $71.4_{\pm0.5}$ | $73.0_{\pm0.5}$ | $54.4_{\pm4.4}$ | $57.1_{\pm1.5}$ | $52.4_{\pm2.8}$ | $\mathbf{68.3}_{\pm1.2}$ | |
| | 1.80% | $59.0_{\pm0.5}$ | $70.6_{\pm0.9}$ | $69.8_{\pm1.1}$ | $72.4_{\pm0.4}$ | $74.3_{\pm0.1}$ | $64.2_{\pm1.7}$ | $66.7_{\pm1.0}$ | $64.3_{\pm1.0}$ | $\mathbf{69.1}_{\pm1.1}$ | $71.7_{\pm0.1}$ |
| | 3.60% | $66.3_{\pm1.5}$ | $69.8_{\pm1.4}$ | $69.4_{\pm1.4}$ | $70.6_{\pm0.7}$ | $73.3_{\pm0.4}$ | $69.8_{\pm1.1}$ | $69.0_{\pm0.1}$ | $69.1_{\pm0.1}$ | $\mathbf{70.5}_{\pm1.1}$ | |
| Cora | 1.30% | $67.3_{\pm1.9}$ | $79.8_{\pm1.2}$ | $75.9_{\pm1.2}$ | $80._{\pm0.4}$ | $82.5_{\pm0.4}$ | $63.6_{\pm3.7}$ | $67.0_{\pm1.3}$ | $64.0_{\pm2.3}$ | $\mathbf{81.1}_{\pm0.7}$ | |
| | 2.60% | $67.6_{\pm3.5}$ | $80.1_{\pm0.6}$ | $75.7_{\pm0.9}$ | $81.7_{\pm0.5}$ | $83.6_{\pm0.3}$ | $72.8_{\pm1.1}$ | $73.4_{\pm1.0}$ | $73.2_{\pm1.2}$ | $\mathbf{81.6}_{\pm0.6}$ | $81.2_{\pm0.2}$ |
| | 5.20% | $67.7_{\pm2.2}$ | $79.3_{\pm0.3}$ | $76.0_{\pm0.3}$ | $81.6_{\pm0.8}$ | $82.8_{\pm0.7}$ | $76.8_{\pm0.1}$ | $76.8_{\pm0.1}$ | $76.7_{\pm0.1}$ | $\mathbf{80.7}_{\pm0.4}$ | |
| Ogbn arxiv | 0.05% | $58.6_{\pm0.4}$ | $59.2_{\pm1.1}$ | $61.3_{\pm0.5}$ | $65.5_{\pm0.7}$ | $65.5_{\pm0.6}$ | $47.1_{\pm3.9}$ | $52.4_{\pm1.8}$ | $47.2_{\pm3.0}$ | $\mathbf{68.0}_{\pm0.1}$ | |
| | 0.25% | $59.9_{\pm0.3}$ | $63.2_{\pm0.3}$ | $64.2_{\pm0.4}$ | $66.1_{\pm0.4}$ | $68.8_{\pm0.2}$ | $57.3_{\pm1.1}$ | $58.6_{\pm1.2}$ | $56.8_{\pm0.8}$ | $\mathbf{69.6}_{\pm0.2}$ | $\mathbf{71.4}_{\pm0.1}$ |
| | 0.50% | $59.5_{\pm0.3}$ | $64.0_{\pm1.4}$ | $63.1_{\pm0.5}$ | $66.8_{\pm0.4}$ | $69.6_{\pm0.2}$ | $60.0_{\pm0.9}$ | $60.4_{\pm0.8}$ | $60.3_{\pm0.4}$ | $\mathbf{69.8}_{\pm0.1}$ | |
| Flickr | 0.10% | $46.3_{\pm0.2}$ | $46.5_{\pm0.4}$ | $45.9_{\pm0.1}$ | $46.6_{\pm0.2}$ | $47.1_{\pm0.1}$ | $41.8_{\pm2.0}$ | $42.5_{\pm1.8}$ | $42.0_{\pm0.7}$ | $\mathbf{45.8}_{\pm0.1}$ | |
| | 0.50% | $45.9_{\pm0.1}$ | $47.1_{\pm0.1}$ | $45.0_{\pm0.1}$ | $47.0_{\pm0.1}$ | $47.0_{\pm0.2}$ | $44.0_{\pm0.4}$ | $43.9_{\pm0.9}$ | $43.2_{\pm0.1}$ | $\mathbf{46.8}_{\pm0.2}$ | $47.2_{\pm0.1}$ |
| | 1.00% | $44.6_{\pm0.1}$ | $47.1_{\pm0.1}$ | $45.0_{\pm0.1}$ | $47.1_{\pm0.1}$ | $47.3_{\pm0.3}$ | $44.6_{\pm0.2}$ | $44.4_{\pm0.6}$ | $44.1_{\pm0.4}$ | $\mathbf{47.3}_{\pm0.1}$ | |
| Reddit | 0.01% | $88.2_{\pm0.2}$ | $88.0_{\pm1.8}$ | $88.4_{\pm0.4}$ | $89.7_{\pm0.2}$ | $91.1_{\pm0.4}$ | $46.1_{\pm4.4}$ | $53.1_{\pm2.5}$ | $46.6_{\pm2.3}$ | $\mathbf{89.2}_{\pm0.2}$ | |
| | 0.20% | $90.5_{\pm1.2}$ | $90.1_{\pm0.5}$ | $88.8_{\pm0.4}$ | $90.3_{\pm0.3}$ | $91.5_{\pm0.4}$ | $66.3_{\pm1.9}$ | $71.0_{\pm1.6}$ | $58.5_{\pm2.1}$ | $\mathbf{90.6}_{\pm0.2}$ | $93.9_{\pm0.0}$ |
| | 3.00% | $90.8_{\pm0.9}$ | OOM | $89.2_{\pm0.2}$ | $91.0_{\pm0.3}$ | $93.7_{\pm0.1}$ | $78.4_{\pm1.3}$ | $81.3_{\pm1.1}$ | $82.2_{\pm1.4}$ | $\mathbf{93.3}_{\pm0.1}$ | |

Table 3: *Standard Supervised Settings for Node Classification:* PLGC consistantly ourperforms the existing self-supervided methods, while achieving similar performance compared to the existing supervised methods even when they are trained using 100% clean labels. (OOM: out-of-memory error.)

**Label Noise and Multi-Source Settings.** To study robustness, we introduce node-level label noise during the condensation stage by randomly corrupting a fraction of node labels. We consider noise levels $\{0.0, 0.3, 0.5, 0.7, 0.9\}$, where 0.0 denotes fully clean labels. For multi-source experiments, each dataset is partitioned into three disjoint subgraphs. In transductive settings, training, validation, and test nodes are distributed across sources, whereas in inductive settings only the training graph is partitioned.

## 5.1 Node Classification

### 5.1.1 Single-Source: Clean-Label Fine-Tuning

We first evaluate node classification under a single-source setting where all node labels are available during fine-tuning. We compare PLGC with graph coarsening (Huang et al., 2021), graph coreset methods (Random, Herding (Welling, 2009), K-Center (Sener & Savarese, 2017), dataset condensation approaches (DC-Graph (Zhao et al., 2020; Jin et al., 2022b)), and state-of-the-art supervised graph condensation methods, including GCond-X, GCond (Jin et al., 2022b), SFGC (Zheng et al., 2023), and GEOM (Zhang et al., 2024).

We refer to methods as supervised when ground-truth labels are used during condensation, whereas self-supervised methods rely solely on intrinsic structural or feature-based signals without access to ground-truth labels. Supervised condensation methods follow a two-stage pipeline: (i) a condensation stage that synthesizes a labeled condensed graph using a GCN, and (ii) supervised training of a GNN on the condensed graph for downstream prediction. Random, Herding, and K-Center correspond to coreset or selection-based approaches that reduce dataset size without learning representations from intrinsic supervisory signals. While these methods do not rely on labels during selection, they are not designed to preserve task-relevant embedding structure for downstream learning. PLGC bridges this gap by learning representations directly from structural and feature-based signals without access to ground-truth labels during condensation, thereby producing an unlabeled condensed graph through a self-supervised objective. PLGC produces an unlabeled condensed graph via self-supervised learning. We therefore fine-tune only the prediction head ($\phi$) using clean labels, while the backbone remains fixed.

Table 3 reports node classification accuracy across five datasets and three compression ratios $r \in (0, 1)$, where the condensed graph size is $N' = rN$. Here, $N$ denotes the total number of nodes in transductive settings and the number of training nodes in inductive settings. PLGC consistently outperforms existing self-supervised methods and achieves performance comparable to supervised condensation approaches under clean-label fine-tuning.

### 5.1.2 Single-Source: Few-Shot Self-Supervised Fine-Tuning

In few-shot settings, only a small number of labeled nodes per class are available during fine-tuning. Here, we consider a 3-shot setup, *i.e.,* the training graph is obtained from a single source for pseudo-label learning, and only three labeled nodes per class are used for downstream adaptation.

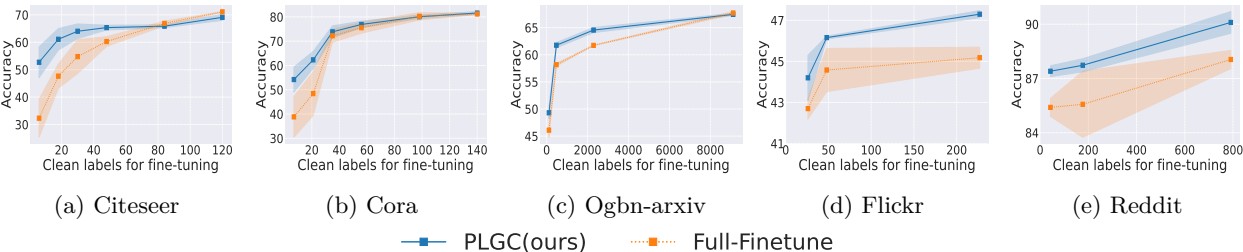

| (a) Citeseer | (b) Cora | (c) Ogbn-arxiv | (d) Flickr | (e) Reddit |

PLGC(ours)   Full-Finetune

Figure 3: *Single-Source with Label Noise for Node classification:* By leveraging the self-supervised condensed graphs, PLGC achieves better performance at lesser number of clean labelled nodes during funetuning stage, and demonstrates faster convergence compared to the 'full-finetune' baselines.

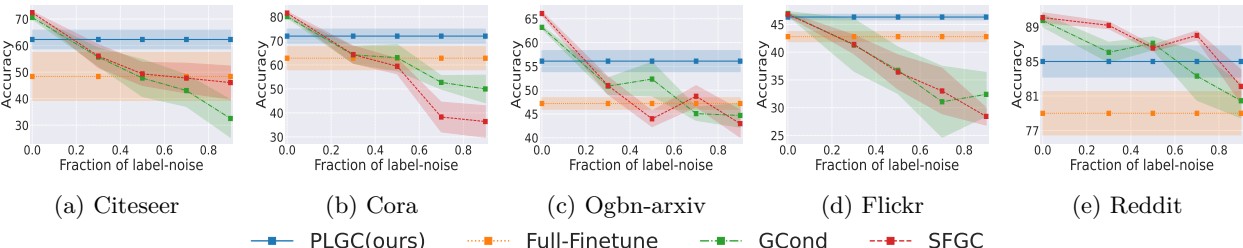

| (a) Citeseer | (b) Cora | (c) Ogbn-arxiv | (d) Flickr | (e) Reddit |

PLGC(ours)   Full-Finetune   GCond   SFGC

Figure 4: *Single-Source with Label Noise for Node classification:* The superiority of our PLGC method is clearly visible as node-level label noises are increased, producing a robust, stable accuracy compared to the supervised GCond and SFGC methods.

Table 4 compares PLGC with supervised GCond and CTGC, the only existing self-supervised condensation method for node classification. PLGC consistently outperforms CTGC for all datasets. Figure 3 further shows the node classification performance as the number of labeled nodes increases. We compare against a *full fine-tune* baseline, where a GNN is trained directly on labeled nodes without condensation. PLGC achieves higher accuracy and faster convergence, demonstrating improved label efficiency and stronger representations.

| 3-Shot Classification | r | GCond | CTGC | PLGC (Ours) | No Condensed Graph |
|---|---|---|---|---|---|
| Cora | 2.60% | 55.6 ± 3.0 | 70.2±2.5 | **73.4 ± 3.2** | 65.9 ± 4.3 |
| Citeseer | 1.80% | 55.9 ± 0.8 | 63.2 ± 1.8 | **63.4 ± 3.7** | 57.2 ± 2.4 |
| Ogbn-Arxiv | 0.05% | 44.6 ± 0.4 | 48.1 ± 0.5 | **48.7 ± 0.3** | 46.3 ± 3.7 |
| Reddit | 0.1% | 70.0 ± 1.2 | 86.4 ± 0.2 | **88.3 ± 0.2** | 85.4 ± 0.3 |

Table 4: Compression with CTGC, the only existing self-supervised graph condensation method for 3-shot classification tasks.

### 5.1.3 Single-Source: Label Noise

We evaluate robustness to label noise under a single-source setting by corrupting a fraction of node labels during condensation. Fine-tuning is performed using a small subset of clean-labeled nodes, with at least one node per class.

For PLGC, the prediction head is randomly initialized and trained solely on clean labels. In contrast,

| Datasets | r | # Clean Labels | # Training Labels |
|---|---|---|---|
| Citeseer | 1.80% | 18 | 120 |
| Cora | 2.60% | 21 | 140 |
| Ogbn-arxiv | 0.05% | 113 | 90,941 |
| Flickr | 1.00% | 26 | 44,625 |
| Reddit | 1.00% | 43 | 153,932 |

Table 5: Compression ratios and available clean nodes for finetuning for 'Single-source with Label noise' setup.

GCond and SFGC reuse their prediction heads trained during condensation and fine-tune them using clean labels. Compression ratios and fine-tuning label counts are summarized in Table 5. Figure 4 shows that supervised condensation methods perform well at 0% noise – but degrade rapidly as noise increases, eventually reaching a saturation point where noisy condensed graphs limit downstream performance. While large datasets (*e.g.,* Reddit) retain sufficient clean labels under moderate noise, this assumption can be unrealistic in practice. In contrast, PLGC remains robust across different noise levels and significantly outperforms supervised methods at higher noise rates, while using only ≈ 1 clean-labeled node per class to fine-tune.

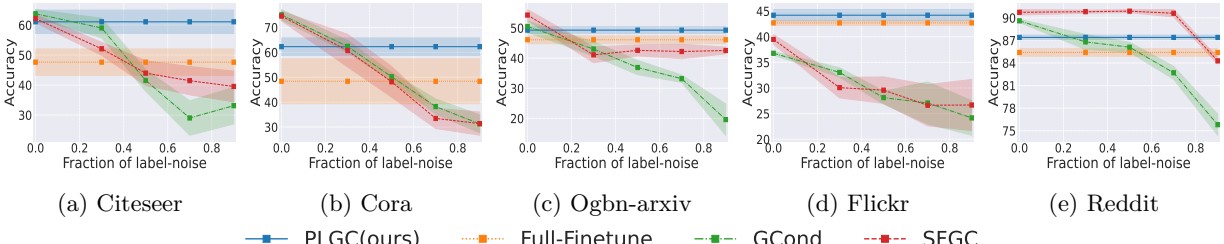

Figure 5: *Multi-Source with Label Noise for Node classification:* The superiority of our PLGC method is clearly visible as node-level label noises are increased, producing a robust, stable accuracy compared to the supervised GCond and SFGC methods.

### 5.1.4 Multi-Source: Label Noise

We extend the evaluation to a multi-source setting with three subgraphs, where label noise is introduced independently within each source during condensation. Fine-tuning uses a small set of clean-labeled nodes pooled across all sources.

Condensed graphs are generated separately from each source and used to train a downstream GNN. PLGC initializes the prediction head randomly and trains it only on clean labels, whereas GCond and SFGC fine-tune their pre-trained heads. Experimental configurations are listed in Table 6.

| Datasets | r | # Clean Labels | # Training Labels |
|---|---|---|---|
| Citeseer | 1.80% | 24 | 120 |
| Cora | 2.60% | 28 | 140 |
| Ogbn-arxiv | 0.05% | 165 | 90,941 |
| Flickr | 1.00% | 33 | 44,625 |
| Reddit | 1.00% | 123 | 153,932 |

Table 6: Compression ratios and clean nodes during fine-tuning for 'Multi-source with Label noise' setup.

Figure 5 shows that supervised methods degrade rapidly as noise increases, particularly on smaller graphs. In contrast, PLGC consistently achieves higher accuracy under high noise, demonstrating strong robustness and generalization in heterogeneous multi-source environments.

## 5.2 Link Prediction

We evaluate condensed graphs on downstream link prediction under a single-source noisy setting. In transductive settings, edges are split into training, validation, and test sets in a 1:1:2 ratio. In inductive settings, the condensed graphs generated for node classification are reused. During fine-tuning, only the prediction head ($\phi$) is updated, while the GNN backbone (GNN$_\theta$) remains frozen. Performance is evaluated using AUROC (Davis & Goadrich, 2006).

Figure 6 compares PLGC with GCond, SFGC, and a *Full Fine-Tune* baseline. PLGC achieves superior performance under increasing label noise for all datasets except Flickr. Notably, both PLGC and supervised condensation methods (under clean labels) outperform full fine-tuning on most datasets, highlighting the importance of node-level optimization even for link prediction. Supervised methods remain sensitive to label noise due to reliance on node-level supervision during condensation, whereas PLGC maintains stable performance through self-supervised learning.

## 5.3 Key Observations and Summary

**(i) Robustness to Label Noise.** PLGC consistently outperforms supervised condensation methods under moderate to high label noise, demonstrating strong robustness to noisy supervision.

**(ii) Label Efficiency.** PLGC achieves competitive performance using only a few labeled nodes per class during fine-tuning, substantially reducing annotation requirements.

**(iii) Multi-Source Generalization.** Across heterogeneous sources, PLGC maintains stable performance, whereas supervised methods degrade rapidly due to inconsistent or noisy labels.

**(iv) Transferability Across Tasks.** Condensed graphs produced by PLGC generalize effectively from node classification to link prediction, underscoring the quality of the learned representations.

Overall, these results demonstrate that PLGC provides a robust and label-efficient framework for graph dataset condensation under noisy and multi-source settings.

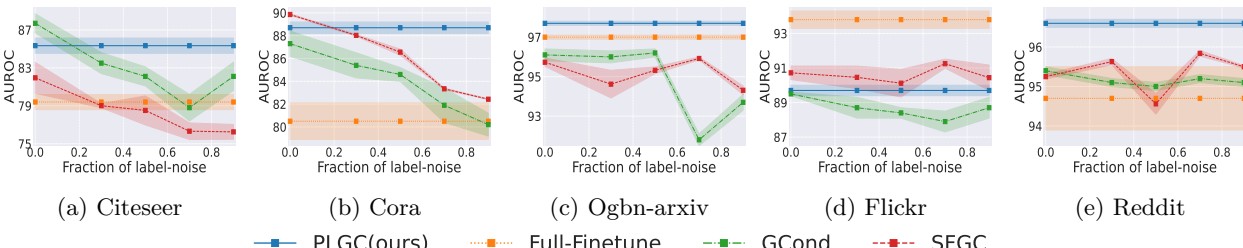

Figure 6: Single-Source with Label Noise for Link Prediction: PLGC achieves stable performance, demonstrating robustness by utilizing structural and semantic information through self-supervised learning and outperforming supervised methods like GCond and SFGC at larger noise level.

## 6 Conclusion

In this work, we introduce *PLGC*, a self-supervised graph condensation framework that eliminates reliance on ground-truth labels by directly constructing pseudo-labels from latent node representations. Unlike prior condensation methods that depend on noisy or scarce supervision, PLGC leverages a principled prototype-based pseudo-labeling mechanism to capture latent cluster structure and guide the condensation process in a fully label-free manner.

We provided theoretical guarantees showing that, under mild distributional assumptions, the learned pseudo-labels preserve essential latent-space properties, including cluster separation and concentration of empirical centroids, with high probability. These results formally justify the stability and reliability of pseudo-label–driven condensation and explain why PLGC remains robust even when supervision is severely corrupted or unavailable.

Extensive experiments on five benchmark datasets demonstrate that PLGC achieves performance comparable to state-of-the-art supervised graph condensation methods in clean-label settings, while substantially outperforming them under increasing levels of label noise. Moreover, PLGC consistently excels in few-shot and multi-source scenarios, highlighting its ability to aggregate information across heterogeneous and noisy data sources-an increasingly common setting in real-world graph applications. Overall, PLGC bridges the gap between self-supervised representation learning and dataset condensation by unifying theoretical rigor with empirical robustness.

However, our approach assumes that meaningful latent clusters can be recovered from node features and neighborhood structure. When connectivity patterns are only *weakly correlated* with feature similarity (e.g., heterophilic graphs or adversarial structures), the learned pseudo-labels may not align with task-relevant semantics, potentially limiting the quality of condensation. Incorporating structural priors or adaptive neighborhood modeling could help mitigate this limitation in such settings.

Extending PLGC to more complex graph settings presents several promising directions for future work. In particular, adapting the framework to heterogeneous graphs would require incorporating type-specific encoders, designing type-aware or shared pseudo-label representations, and developing assignment mechanisms that account for cross-type interactions. Ensuring consistent alignment across heterogeneous node and edge types remains a key challenge for improving the generality of pseudo-label–driven condensation. *More broadly, integrating robustness to structural heterogeneity and distributional shift is an important direction for enhancing the applicability of PLGC.* Additional directions include extending PLGC to dynamic graphs, deriving tighter theoretical bounds under weaker assumptions, and applying the framework to continual and privacy-preserving graph learning settings.

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

# A Theoretical Analysis of Pseudo-Labeled Graph Condensation

This appendix provides a complete theoretical analysis of why the pseudo-labels $\widetilde{Y} = \{\widetilde{y}_1, \ldots, \widetilde{y}_K\}$ produced by PLGC behave as assignment-weighted centroids and concentrate around the true latent centers. Our main theorem shows that: (1) each pseudo-label $\widetilde{y}_k$ concentrates around the true latent center $\mu_k$, and (2) the pseudo-labels preserve cluster separation, ensuring that the condensed graph encodes the same latent geometric structure as the original graph. This result explains why PLGC produces stable and robust condensed graphs, especially under label noise or weak supervision. We also demonstrate the sample complexity to ensure stronger separation among the learned pseudo-labels. In the following, we first present a set of supporting lemmas followed by the main theorem for our paper.

**Lemma 1** (Existence of a 1/2-net on unit sphere). *Let $\mathbb{S}^{d-1} := \{u \in \mathbb{R}^d : \|u\|_2 = 1\}$ denote the unit sphere. There exists a finite set $\mathcal{N} \subset \mathbb{S}^{d-1}$ such that for every $u \in \mathbb{S}^{d-1}$, there exists $v \in \mathcal{N}$ satisfying $\|u - v\| \leq \frac{1}{2}$. Moreover, the cardinality of $\mathcal{N}$ satisfies $|\mathcal{N}| \leq 5^d$.*

*Proof.* Construct $\mathcal{N}$ as any maximal 1/2–separated subset of $\mathbb{S}^{d-1}$; that is, choose points on the sphere greedily so that the distance between any two chosen points is $> 1/2$, and stop when no further point can be added. By maximality the chosen set is automatically a 1/2-net: if there existed $u \in \mathbb{S}^{d-1}$ with $\|u-v\| > 1/2$ for all chosen $v$, then $u$ could be added, contradicting maximality.

It remains to bound the cardinality. For each $v \in \mathcal{N}$ consider the Euclidean ball $B(v, r) \subset \mathbb{R}^d$ of radius $r = \frac{1}{4}$. Because the points in $\mathcal{N}$ are 1/2-separated, the balls $B(v, \frac{1}{4})$ are pairwise disjoint. Moreover, every such ball is contained in the Euclidean ball centered at the origin of radius $1 + \frac{1}{4} = \frac{5}{4}$, since each $v \in \mathbb{S}^{d-1}$ has $\|v\| = 1$. Hence the union of these disjoint balls is contained in $B(0, 5/4)$. Taking Lebesgue volumes (and denoting by $\text{vol}(B(0, r))$ the volume of a radius-$r$ ball in $\mathbb{R}^d$) we obtain

$$|\mathcal{N}| \cdot \text{vol}\big(B(0, \tfrac{1}{4})\big) \leq \text{vol}\big(B(0, \tfrac{5}{4})\big).$$

Using the scaling of Euclidean volume, $\text{vol}(B(0, r)) = r^d \text{vol}(B(0, 1))$, this yields

$$|\mathcal{N}| \leq \frac{(5/4)^d \, \text{vol}(B(0, 1))}{(1/4)^d \, \text{vol}(B(0, 1))} = \Big(\frac{5/4}{1/4}\Big)^d = 5^d.$$

$\square$

**Lemma 2** (Directional concentration via Chernoff). *Let $\{z_i\}_{i=1}^n$ be the set of independent samples corresponding to $k$-th pseudo-label with true latent centers, $\mu_k$. We denote $q_{ik} \in \{0, 1\}$ as the label assignments and define $s_k := \sum_{i=1}^n q_{ik}$. Assume that for every unit vector $u \in \mathbb{S}^{d-1}$, the scalar random variable $u^\top(z_i - \mu_k)$ is $\sigma$-sub-Gaussian, i.e.,*

$$\mathbb{E}\big[\exp(\lambda u^\top(z_i - \mu_k))\big] \leq \exp\Big(\frac{\sigma^2 \lambda^2}{2}\Big) \qquad \forall \lambda \in \mathbb{R}.$$

*Then for any $t > 0$,*

$$\Pr\Big(\Big|u^\top\Big(\frac{\sum_i q_{ik} z_i}{s_k} - \mu_k\Big)\Big| \geq t\Big) \leq 2 \exp\Big(-\frac{s_k t^2}{2\sigma^2}\Big).$$

*Proof.* We fix $u \in \mathbb{S}^{d-1}$ and define $X_i := u^\top(z_i - \mu_k)$, and $S := \sum_{i=1}^n q_{ik} X_i$. Then $u^\top\Big(\frac{\sum_i q_{ik} z_i}{s_k} - \mu_k\Big) = \frac{S}{s_k}$.

Since each $X_i$ is $\sigma$-sub-Gaussian and $q_{ik} \in \{0, 1\}$, we have for any $\lambda > 0$,

$$\mathbb{E}[\exp(\lambda q_{ik} X_i)] \leq \exp\Big(\frac{\sigma^2 \lambda^2 q_{ik}^2}{2}\Big) \leq \exp\Big(\frac{\sigma^2 \lambda^2 q_{ik}}{2}\Big).$$

Given $z_i$'s and therefore, $X_i$'s are independent, we have

$$\mathbb{E}[e^{\lambda S}] = \prod_{i=1}^n \mathbb{E}[e^{\lambda q_{ik} X_i}] \leq \exp\Big(\frac{\sigma^2 \lambda^2}{2} \sum_{i=1}^n q_{ik}\Big) = \exp\Big(\frac{\sigma^2 \lambda^2 s_k}{2}\Big).$$

Next, we apply Markov: $\Pr(S \geq s_k t) = \Pr(e^{\lambda S} \geq e^{\lambda s_k t}) \leq e^{-\lambda s_k t} \mathbb{E}[e^{\lambda S}] \leq \exp\left(-\lambda s_k t + \frac{\sigma^2 \lambda^2 s_k}{2}\right)$.

Since, the above equation is true for any $\lambda \in \mathbb{R}$, we select $\lambda = t/\sigma^2$ to minimize the LHS, yielding

$$\Pr(S \geq s_k t) \leq \exp\left(-\frac{s_k t^2}{2\sigma^2}\right) \iff \Pr\left(\frac{S}{s_k} \geq t\right) = \Pr\left(u^\top \frac{\sum_i q_{ik} z_i}{s_k} - \mu_k \geq t\right) \leq \exp\left(-\frac{s_k t^2}{2\sigma^2}\right).$$

Finally, applying the same argument to $-S$ (or equivalently to $-u$) yields the two-sided bound as claimed in the lemma:

$$\Pr\left(\left|u^\top \left(\frac{\sum_i q_{ik} z_i}{s_k} - \mu_k\right)\right| \geq t\right) \leq 2 \exp\left(-\frac{s_k t^2}{2\sigma^2}\right).$$

$\square$

Next, we analyze the stationarity of our pseudo-label learning objective. Note that the actual loss objective (Eq. 6) enforces cross-view consistency by matching pseudo-label assignments of one augmented view with embeddings of another. Although this cross-view formulation is crucial for effectively learning the pseudo-labels, the optimization admits a simpler characterization with respect to the pseudo-label variables.

Specifically, when considering fixed embeddings and assignment variables, the objective function reduces to aggregating the similarity between the node embeddings and their assigned pseudo-labels. Therefore, without loss of generality, we analyze the stationary points of the following simplified pseudo-label learning task and show that, under the first-order optimality conditions, pseudo-labels align with the assignment-weighted sum of embeddings.

**Lemma 3** (Stationarity Condition of pseudo-labels). *Consider the following simplified pseudo-label learning objective associated with Eq. 6:*

$$\min_{\{\widetilde{y}_k\}_{k=1}^K} -\sum_{i=1}^n \sum_{k=1}^K q_{ik} z_i^\top \widetilde{y}_k \quad \textit{subject to } \|\widetilde{y}_k\|_2 = 1 \ \ \forall k.$$

*Any stationary point $\{\widetilde{y}_k\}_{k=1}^K$ of this problem satisfies, for each $k$, $\widetilde{y}_k \ \| \ \sum_{i=1}^n q_{ik} z_i$ or equivalently, $\widetilde{y}_k = \frac{\sum_{i=1}^n q_{ik} z_i}{\left\| \sum_{i=1}^n q_{ik} z_i \right\|}$.*

*Proof.* By introduce Lagrange multipliers $\lambda_k \in \mathbb{R}$ for the unit-norm constraints, we get:

$$\mathcal{L} = -\sum_{i=1}^n \sum_{k=1}^K q_{ik} z_i^\top \widetilde{y}_k + \sum_{k=1}^K \lambda_k \left(\|\widetilde{y}_k\|_2^2 - 1\right).$$

At a stationary point, the first-order optimality (KKT) conditions require: $\nabla_{\widetilde{y}_k} \mathcal{L} = 0 \quad \forall k$.

Computing the gradient yields: $-\sum_{i=1}^n q_{ik} z_i + 2\lambda_k \widetilde{y}_k = 0$. Finally, we can rearrange to obtain $\widetilde{y}_k = \frac{1}{2\lambda_k} \sum_{i=1}^n q_{ik} z_i$. Therefore, $\widetilde{y}_k$ is collinear with $\sum_{i=1}^n q_{ik} z_i$. Imposing the constraint $\|\widetilde{y}_k\|_2 = 1$ yields the stated normalized form. $\square$

**Theorem 1** (Pseudo-labels Concentration and Interior Point recovery). *Let $\{z_i\}_{i=1}^n$ be the independent samples corresponding to $K$ latent centers, with pseudo-label assignments denoted by $q_{ik} \in \{0,1\}$. The samples $\{z_i : q_{ik} = 1\}$ are corresponding to pseudo-label $k$. We denote $\mu_k$ be the true latent centers (or the population centers) and define $s_k := \sum_{i=1}^n q_{ik}$. We assume:*

**(A1) Sub-Gaussian noise.** *for every unit vector $u \in \mathbb{S}^{d-1}$, the scalar random variable $u^\top (z_i - \mu_k)$ is $\sigma$-sub-Gaussian, i.e., $\mathbb{E}\left[\exp(\lambda u^\top (z_i - \mu_k))\right] \leq \exp\left(\frac{\sigma^2 \lambda^2}{2}\right) \quad \forall \lambda \in \mathbb{R}$.*

**(A2) Separation.** *The true latent centers are well-separated i.e., $\Delta := \min_{j \neq k} \|\mu_j - \mu_k\| > 0$.*

*We set the deviation parameter $\epsilon_k := 4\sigma\sqrt{\frac{d+\log(2K/\delta)}{s_k}}$. Given that the pseudo-labels satisfies the stationarity condition as in Lemma 3, the following statements hold simultaneously for all k with probability at least $1-\delta$:*

(i) **Pseudo-label Concentration.** *Each pseudo-label satisfies $\|\widetilde{y}_k - \mu_k\| \leq \epsilon_k$, i.e., pseudo-labels produced by PLGC remains tightly concentrated to the true latent centers.*

(ii) **Interior-point recovery.** *Any point $z_i$ assigned to k-th pseudo-label (i.e., $q_{ik} = 1$) such that $\|z_i - \mu_k\| < \frac{\Delta}{2} - \epsilon_k$ remains closer to $\widetilde{y}_k$ than other labels $\widetilde{y}_\ell$ – indicating that, the interior points are always correctly assigned.*

(iii) **Sample Complexity & Pseudo-Label Separation.** *A sufficient sample size to ensure a positive interior margin of $\epsilon_{\max} \leq \Delta/\beta$ is: $s_k \geq \frac{16\,\sigma^2\,\beta^2}{\Delta^2}\left(d+\log(2K/\delta)\right)$, and leads to a stronger separation of pseudo-labels: $\|\widetilde{y}_k - \widetilde{y}_\ell\| \geq (1 - \frac{2}{\beta})\Delta \quad \forall k \neq \ell$*

*Proof.* We prove each claim in order. The argument uses the standard net + Chernoff method for sub-Gaussian vectors; the constants above are conservative but explicit.

**(i) Concentration of sample centroids.** Fix a cluster $k$ and a unit vector $u \in \mathbb{S}^{d-1}$. By the sub-Gaussian assumption and directional concentration argument (Lemma 2), we get,

$$\Pr\left(\left|u^\top\left(\frac{\sum_i q_{ik}z_i}{s_k} - \mu_k\right)\right| \geq t\right) \leq 2\exp\left(-\frac{s_k t^2}{2\sigma^2}\right) \qquad \forall t > 0$$

Denoting $\mathcal{N}$ be a 1/2-net of $\mathbb{S}^{d-1}$, we get $|\mathcal{N}| \leq 5^d$ (Lemma 1). Applying the tail bound to each $v \in \mathcal{N}$ and using union bound argument, we yield

$$\Pr\left(\exists v \in \mathcal{N} : \left|v^\top\left(\frac{\sum_i q_{ik}z_i}{s_k} - \mu_k\right)\right| \geq t\right) \leq \sum_{v \in \mathcal{N}} \Pr\left(\left|v^\top\left(\frac{\sum_i q_{ik}z_i}{s_k} - \mu_k\right)\right| \geq t\right) = 2 \cdot 5^d \exp\left(-\frac{s_k t^2}{2\sigma^2}\right).$$

If the maximum error over the net is $< t$, i.e., $\max_{v \in \mathcal{N}} |v^\top(\bar{z}_k - \mu_k)| < t$, then for any $u \in \mathbb{S}^{d-1}$, we can choose $v \in \mathcal{N}$ such that $\|u - v\| \leq 1/2$. Now, we using triangle inequality and decomposition $u = v + (u - v)$, we get:

$$\left|u^\top\left(\frac{\sum_i q_{ik}z_i}{s_k} - \mu_k\right)\right| \leq \left|v^\top\left(\frac{\sum_i q_{ik}z_i}{s_k} - \mu_k\right)\right| + \left|(u - v)^\top\left(\frac{\sum_i q_{ik}z_i}{s_k} - \mu_k\right)\right|.$$

Applying the Cauchy-Schwarz inequality, $\left|(u - v)^\top\left(\frac{\sum_i q_{ik}z_i}{s_k} - \mu_k\right)\right| \leq \|u - v\|\left\|\frac{\sum_i q_{ik}z_i}{s_k} - \mu_k\right\|$, along with the net property $\|u - v\| \leq 1/2$ and the assumption that $|v^\top(\bar{z}_k - \mu_k)| < t$, we obtain:

$$\left|u^\top\left(\frac{\sum_i q_{ik}z_i}{s_k} - \mu_k\right)\right| < t + \frac{1}{2}\left\|\frac{\sum_i q_{ik}z_i}{s_k} - \mu_k\right\|.$$

Taking the supremum over all $u \in \mathbb{S}^{d-1}$ yields the norm $\sup_u \left|u^\top\left(\frac{\sum_i q_{ik}z_i}{s_k} - \mu_k\right)\right| = \left\|\frac{\sum_i q_{ik}z_i}{s_k} - \mu_k\right\|$, leading to the inequality:

$$\left\|\frac{\sum_i q_{ik}z_i}{s_k} - \mu_k\right\| < t + \left\|\frac{\sum_i q_{ik}z_i}{s_k} - \mu_k\right\|.$$

Rearranging the terms, we get: $\left\|\frac{\sum_i q_{ik}z_i}{s_k} - \mu_k\right\| < 2t$. Consequently, the event $\left\{\left\|\frac{\sum_i q_{ik}z_i}{s_k} - \mu_k\right\| \geq 2t\right\}$ is contained within the event that the net bound fails, leading to the final lifting inequality:

$$\Pr\left(\left\|\frac{\sum_i q_{ik}z_i}{s_k} - \mu_k\right\| \geq 2t\right) \leq 2 \cdot 5^d \exp\left(-\frac{s_k t^2}{2\sigma^2}\right).$$

Choosing $t = \sigma\sqrt{\frac{2(d\log 5 + \log(2K/\delta))}{s_k}}$ (i.e., $2t = \epsilon_k$), we get $\Pr(\|\bar{z}_k - \mu_k\| \geq \epsilon_k) \leq \delta/(2K)$. A union bound over all pseudo-labels, $k = 1, \ldots, K$ yields claim-(i) with probability at least $1 - \delta/2$. (Absorb the remaining small failure probability to get overall $1 - \delta$.)

**(ii) Interior-point recovery.** Fix $k$ and a point $z_i$ with $q_{ik} = 1$ and $\|z_i - \mu_k\| < \frac{\Delta}{2} - \epsilon_k$. For any other $\ell \neq k$, we use the triangle inequality:

$$\|z_i - \widetilde{y}_k\| \leq \|z_i - \mu_k\| + \|\mu_k - \widetilde{y}_k\| < (\tfrac{\Delta}{2} - \epsilon_k) + \epsilon_k = \tfrac{\Delta}{2}.$$

On the other hand, applying the triangle inequality and (i),

$$\|z_i - \widetilde{y}_\ell\| \geq \|\mu_k - \mu_\ell\| - \|z_i - \mu_k\| - \|\widetilde{y}_\ell - \mu_\ell\| > \Delta - (\tfrac{\Delta}{2} - \epsilon_k) - \epsilon_k = \tfrac{\Delta}{2}.$$

Therefore, we always have $\|z_i - \widetilde{y}_k\| < \|z_i - \widetilde{y}_\ell\|$ for all $\ell \neq k$. Hence, $z_i$ is correctly assigned to $\widetilde{y}_k$.

**(iii) Sample Complexity & Pseudo-Label Separation.** The sufficient sample size to ensure $\epsilon_{\max} \leq \Delta/\beta$ is obtained by rearranging the expression of $\epsilon_k := 4\sigma\sqrt{\frac{d + \log(2K/\delta)}{s_k}}$ i.e.,

$$s_k \geq \frac{16\,\sigma^2\,\beta^2}{\Delta^2}\left(d + \log(2K/\delta)\right) \qquad \text{for all } k.$$

Finally, $\epsilon_{\max} \leq \Delta/\beta$ naturally leads to a stronger separation bound of pseudo-labels by using triangle inequality as follows:

$$\|\widetilde{y}_l - \widetilde{y}_k\| \geq \|\mu_k - \mu_\ell\| - \|\widetilde{y}_k - \mu_k\| - \|\widetilde{y}_\ell - \mu_\ell\| \geq \Delta - \epsilon_k - \epsilon_\ell \geq \Delta - 2\epsilon_{\max} = \left(1 - \frac{2}{\beta}\right)\Delta$$

$\square$

| Method | Dominant condensation-stage time complexity |
|---|---|
| Herding | $O(mN^2)$ |
| K-Center | $O(mN^2)$ |
| GCond | $O\Big(T(LEd + LNd^2 + Lm^2d + m^2d^2)\Big)$ |
| SFGC | $O\Big(T(LEd + LNd^2 + Lmd^2 + Lmd)\Big)$ |
| GEOM | $O\Big(T(Lmd^2 + Lmd) + M_e r_n^L N d^2\Big)$ |
| ST-GCond | $O\Big((s+1)(Lm^2d + Lmd + m^2d^2 + Nmr^2 + Nm) + k(LEd + LNd^2)\Big)$ |
| CTGC | $O\Big(LEd + N(p+1)d + mN^2 + mNdq + mpd + Lm^2d\Big)$ |
| **PLGC (ours)** | $O\Big(M_s T(LEd + LNd^2 + NKd + Lmd^2)\Big)$ |

Table 7: Computational complexity comparison of representative graph condensation methods for node-level tasks. For compactness, we use $m$ to denote the number of condensed nodes throughout; thus, the class-budget notation in coreset baselines is absorbed into $m$. Here, $N$ and $E$ denote the number of nodes and edges in the original graph, $M_s$ the number of source graphs in PLGC, $K$ the number of pseudo-labels/prototypes, $d$ the embedding dimension, $L$ the number of GNN layers, and $T$ the number of condensation iterations. In addition, $M_e$ denotes the number of expert trajectories in GEOM, $r_n$ the sampled-neighbor factor in GEOM, $s$ the number of sampled subtasks in ST-GCond, $k$ the number of self-supervised teacher tasks in ST-GCond, $r$ the kernel-ridge regression rank in ST-GCond, and $p, q$ the polynomial order and K-means iterations in CTGC, respectively.

## B Computational and Memory Efficiency

Table 7 reports the dominant cost of the *condensation stage*, rather than downstream fine-tuning. For compactness, we summarize the dominant-order terms and use $m$ consistently for the condensed graph size.

| Label -Noise | PLGC: condensation with supervised signals | | | PLGC (self-supervised) |
| | 10% | 25% | 50% | -NA- |
|---|---|---|---|---|
| CORA (r = 2.6%) | 76.4 ± 0.9 | 72.5 ± 2.6 | 70.0 ± 3.4 | 73.4 ± 3.4 |
| CiteSeer (r = 1.8%) | 65.7 ± 1.8 | 65.6 ± 2.6 | 60.1 ± 5.7 | 63.4 ± 3.7 |

Table 8: Comparing performance of PLGC models as we learn condensed graphs by supervised labels in presence of varying label noises for a 3-shot classification task.

Existing graph condensation methods such as GCond (Jin et al., 2022b), SFGC (Zheng et al., 2023), GEOM (Zhang et al., 2024), and ST-GCond (Yang et al., 2025) rely on gradient, trajectory, or meta-matching style optimization, which introduces substantial iterative overhead during condensed graph construction. In particular, ST-GCond further incorporates auxiliary self-supervised teacher tasks to improve transferability, which adds extra computation on the original graph.

CTGC (Gao et al., 2025) is also a relevant recent self-supervised baseline for node-level tasks. However, its complexity includes additional spectral and clustering costs due to structural relay learning, eigenvalue decomposition, and alternating graph generation. As a result, beyond the relay-model training cost, its dominant burden comes from eigendecomposition and clustering on the original graph.

In contrast, PLGC constructs the condensed graph through a closed loop of (i) backbone encoding of the original graph under a constant number of augmentations, (ii) prototype-based pseudo-label assignment, and (iii) condensed-node refinement. This yields an overall complexity of

$$O\big(M_sT(LEd + LNd^2 + NKd + Lmd^2)\big).$$

Since $m \ll N$ and typically $K \ll N$, the additional pseudo-labeling and condensed-node optimization terms are much smaller than processing the original graph.

**Memory Footprint.** Algorithmically, PLGC also avoids storing expert trajectories as in trajectory-matching methods and does not require explicit eigen-decomposition of the original graph as in CTGC. GCond incurs online gradient-matching memory, while SFGC and GEOM rely on offline expert trajectories or checkpoints; ST-GCond further maintains auxiliary self-supervised teacher states. CTGC removes expert-trajectory storage, but its structural branch still requires spectral quantities and clustering intermediates during eigendecomposition and graph generation. In contrast, PLGC only maintains the current backbone parameters, the prototype matrix, the condensed node features, and standard GNN activations during condensation. This makes PLGC attractive for large-scale and multi-source condensation settings, where both computation and memory efficiency are important.

## C  Additional Ablation Study

### C.1  Incorporating Supervision to learn the condensed graph

Our current formulation of PLGC is strictly self-supervised and does not explicitly incorporate node signals. Our design choice is intentional: PLGC is motivated by settings where labels are scarce, noisy, or unreliable, and where supervised condensation objectives can fail or even propagate errors. Therefore, PLGC relies on latent pseudo-labels that preserve the graph's structural and feature statistics, with theoretical guarantees of their stability and alignment.

However, an easy way of incorporating supervised node labels is to learn our condensed graph in a multi-task fashion. As in Figure 2, the backbone GNN model produces node-embeddings that are attached to our proposed self-supervised loss. In presence of supervision, we can include an additional supervised branch along with cross-entropy loss for node classification.

Table 8 presents the results as we incorporate node classification tasks in presence of different levels of label-noises while producing our condensed graphs. We consider 3-Shot Classification task. We observe the benefits of introducing such supervised loss in presence of lower label-noises. However, as higher label

| $K$ | 35 | 70 | 140 |
|---|---|---|---|
| Cora | $81.1 \pm 0.7$ | $81.6 \pm 0.6$ | $80.7 \pm 0.4$ |
| CiteSeer | $68.3 \pm 1.2$ | $69.1 \pm 1.1$ | $70.5 \pm 1.1$ |

Table 9: Comparing the performance as we vary the number of pseud-labels, $K$.

| | Cora | | | CiteSeer | | |
|---|---|---|---|---|---|---|
| K | 35 | 70 | 140 | 35 | 70 | 140 |
| NMI | 0.54 | 0.55 | 0.53 | 0.41 | 0.43 | 0.44 |
| ARI | 0.46 | 0.47 | 0.44 | 0.39 | 0.45 | 0.42 |
| Silhouette | 0.24 | 0.22 | 0.18 | 0.21 | 0.21 | 0.19 |

Table 10: NMI, ARI and Shilhouette scores for test node embeddings, obtained from reconstructed GNN backbone models with varying $K$ for Cora and CiteSeer before supervised finetuning.

noises are introduced, the condensed graph often becomes unstable, producing higher variance and often poor results compared to our original self-supervised formulation.

## C.2 Sensitivity of the number of pseudo-labels ($K$)

In PLGC, the number of pseudo-labels, $K$ directly controls the size of the condensed graph (since $N' = K$) and plays a role analogous to the number of clusters in latent space. Therefore, we fix $K$ via standard compression ratios, $r$. Intuitively, smaller $K$ may under-represent complex structures, while larger $K$ increases fidelity but reduces compression. Importantly, our theoretical results suggest that, given sufficient sample size per cluster, pseudo-labels remain well-separated and concentrated (Section 4), which provides some robustness to moderate variations in $K$. In the following, we present the scores as we increase the number of nodes for Cora and CiteSeer (for standard settings without any label noise): We observe a clear improvement of performance as we increase $K$ for CiteSeer while observing performance saturatation for Cora.

## C.3 Quality of self-supervised condensation with varying $K$

Next, we investigate the quality of node embeddings, obtained from reconstructed GNN backbone models using our self-supervised condensed graphs without supervised finetuning. We vary the number of prototypes $K$ to assess the quality of pseudo-labels and, consequently, the effectiveness of our self-supervised graph condensation. We evaluate the learned embeddings both visually using t-SNE (see Figure 7) and quantitatively via NMI, ARI, and Silhouette scores on test nodes (see Table 10). The results show that performance remains relatively stable across different values of $K$, indicating that the learned representations are robust to the choice of prototype granularity.

A moderate number of prototypes (e.g., $K = 70$) provide slightly better alignment with ground-truth labels, while further increasing $K$ does not produce consistent improvements and may degrade cluster compactness, as reflected by lower Silhouette scores. This suggests that excessively large $K$ often also leads to over-fragmentation without meaningful gains, and a moderate value strikes a good balance between capturing structure and maintaining coherent embeddings.

## C.4 Hyper-parameter sensitivities for condensed graph optimization step

Next, we analyse the sensitivity of two important hyper-parameters for learning the condensed graphs. As we can see in Equation 6, condensed graph learning is a 2-step iterative process of swapped-assignment view prediction, $\ell_{swav}$ (Equation 4) and balanced node assignment loss, $\ell_{assign}$ (Equation 5).

For swapped-assignment view prediction, $\ell_{swav}$, temperature scale, $\tau$ is an important hyperparameter that effectively controls the sharpness of the similarity scores between node embeddings and pseudo-labels. Table 11 presents the performance of our PLGC method as we vary temperature for standard node classification

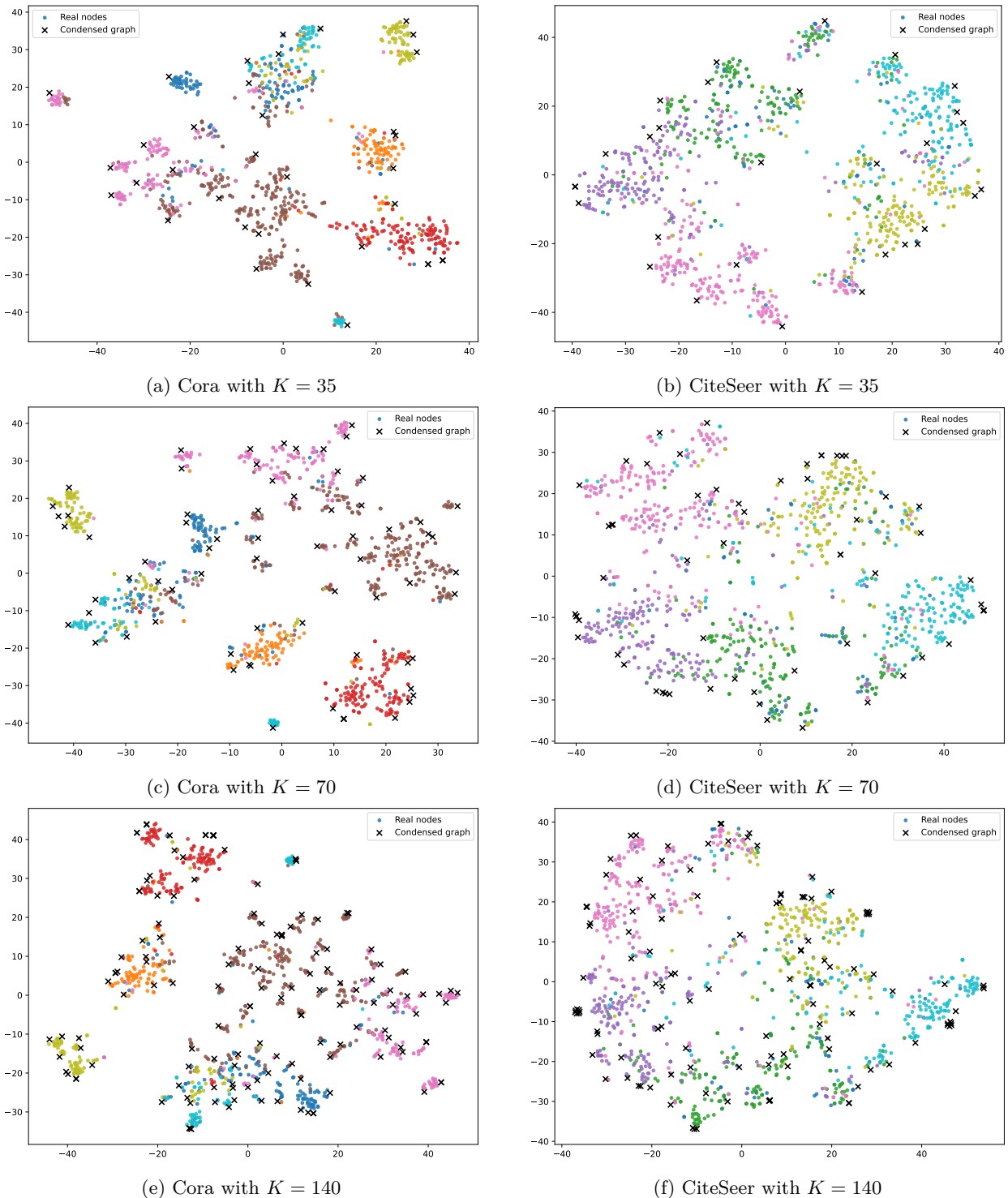

Figure 7: t-SNE visualization of test node embeddings as we vary $K$ for Cora and CiteSeer, obtained from reconstructed GNN backbone models before supervised finetuning. Nodes from different classes are shown in different colors and while pseudo-labels of our PLGC condensed graphs are shown using 'X' symbols.

| Temperature | 0.01 | **0.1** | 1 | 10 |
|---|---|---|---|---|
| Cora | $66.6 \pm 0.1$ | $\mathbf{81.6 \pm 0.6}$ | $79.4 \pm 1.6$ | $75.5 \pm 1.2$ |
| CiteSeer | $50.7 \pm 0.4$ | $\mathbf{69.1 \pm 1.1}$ | $67.7 \pm 1.0$ | $67.5 \pm 1.3$ |

Table 11: Sensitivity analysis of temperature scale for learning the condensed graphs.

| Sinkhorn-Iterations | 1 | 5 | 10 | 15 |
|---|---|---|---|---|
| Cora | $79.7 \pm 0.9$ | $\mathbf{81.6 \pm 0.6}$ | $81.3 \pm 1.3$ | $80.8 \pm 0.8$ |
| CiteSeer | $65.2 \pm 1.8$ | $\mathbf{69.1 \pm 1.1}$ | $68.9 \pm 0.6$ | $68.4 \pm 1.6$ |

Table 12: Sensitivity analysis of sinkhorn-iterations while learning the condensed graphs.

settings with all clean labels for fine-tuning. We observe that we achieve the best performance at $\tau = 0.1$ and the performance degrades as we increase or decrease $\tau$.

The balanced node assignment loss, $\ell_{assign}$ applies iterative Sinkhorn-Knopp algorithm to equally distribute the node embeddings among the pseudo-labels. Table 12 presents the performance as we vary the number of iterations for the Sinkhorn-Knopp algorithm, denoted as Sinkhorn-Iterations. Our results suggest that Sinkhorn-Knopp algorithm converges with 5 iterations and does not change much as we increase the number of iterations.

