# OpenReview forum: "PLGC: Pseudo-Labeled Graph Condensation"
_TMLR — Accepted by TMLR_

### Review · Reviewer_R4Mr · 2026-03-26

**Summary Of Contributions:**

The paper introduces a self-supervised framework that synthesizes a compact synthetic graph from a large dataset without requiring ground-truth labels. The proposed method uses geometry preserving distribution alignment and learns latent prototypes to get node pseudo-labels. This approach effectively addresses the supervision bottleneck and can be used for multi-source graph data. Ultimately, it allows for training high-performance GNNs on a fraction of the original data, significantly reducing the computational overhead of large-scale graph learning.

A key weakness of PLGC is its strictly self-supervised nature, which lacks a mechanism to incorporate partial ground-truth labels when available; this label-blind objective may prioritize structural patterns that are irrelevant to the specific downstream task. Furthermore, it is not clear how the method would behave depending on the choice of K (predefined number of latent prototypes).

**Additional Comments:**

N/A

**Audience:**

Yes

**Audience Explanation:**

Yes, as the method proposes a scalable, label-free solution to the data bottleneck in large-scale Graph Neural Network training. The ability to condense unlabeled and multi-source graphs provides significant utility for research and applications.

**Claims And Evidence:**

Yes

**Claims Explanation:**

The claims are well-supported by a combination of rigorous mathematical proofs and extensive empirical benchmarks across different datasets. The results convincingly demonstrate that the proposed method can condense graphs to achieve significant performance gains in the absence of ground-truth labels.

**Requested Changes:**

It would be great if the authors could identify potential failure cases, specifically when node connectivity does not align with cluster feature similarity. Additionally, providing a detailed sensitivity analysis for $K$ is essential to understand how this impacts the synthetic graph's representation across different datasets.

---

### Review · Reviewer_f1pB · 2026-04-19

**Summary Of Contributions:**

Strength:
1. PLGC introduces a self-supervised framework that eliminates the need for ground-truth labels during the condensation process.
2. By circumventing the reliance on ground-truth labels, PLGC is inherently more robust to label noise, scarcity, and distribution shifts.
3. PLGC is supported by theoretical analyses that provide guarantees on the stability of pseudo-labels and the accuracy of embedding alignment.

Weakness:
1. The core mechanism of using pseudo-labels and alternating optimization for graph condensation represents an incremental improvement rather than a paradigm shift, building upon existing self-supervised learning and clustering techniques.
2. The paper's comparison against the most recent relevant works like CTGC (2025) and SGDC (2024) is not exhaustive. While CTGC is directly compared experimentally (with perceived outperformance), SGDC is primarily discussed theoretically in the related work section, lacking direct comparative experimental results to unequivocally establish PLGC's superiority in its intended domain. The scope of experimental validation for "self-supervised" baselines is also somewhat ambiguous (Table 3).
3. The theoretical analysis relies on strong assumptions (e.g., sub-Gaussian latent structure, separability) that may not hold. Furthermore, the connection between the theoretical guarantees of pseudo-label quality (concentration, separation) and the actual performance improvement in downstream tasks remains largely indirect, lacking a strong theoretical bridge.
4. While PLGC aims to preserve "latent structure," it's unclear if these self-generated pseudo-labels inherently capture the task-relevant discriminative signals needed for specific downstream tasks, or if they primarily reflect general community structure.

**Audience:**

Yes

**Audience Explanation:**

PlGC offers a practical, robust, and theoretically-motivated solution that is likely to be of interest to researchers in the field.

**Claims And Evidence:**

Yes

**Claims Explanation:**

The paper makes several strong claims regarding PLGC's effectiveness and robustness. Most of them are well-supported. However, the evidence could be clearer and more convincing in hyperparameter sensitivity and experimental clarity.

**Requested Changes:**

See the weakness.

---

### Review · Reviewer_z5YG · 2026-04-22

**Summary Of Contributions:**

This paper proposes Pseudo-Labeled Graph Condensation (PLGC), a self-supervised framework for graph dataset condensation that eliminates the reliance on ground-truth labels. Existing graph condensation methods typically require clean supervised labels to guide gradient or representation matching, making them fragile in real-world scenarios where labels are scarce, noisy, or subject to distribution shifts. PLGC addresses this by jointly learning latent pseudo-labels (prototypes) and node assignments through an alternating optimization procedure. Specifically, it employs a swapped-assignment view prediction loss and a balanced assignment loss (via Sinkhorn-Knopp normalization) to infer stable pseudo-labels from node embeddings. These pseudo-labels then guide the construction of a condensed synthetic graph by minimizing the divergence between the condensed graph’s embeddings and the pseudo-labels. The key contributions are:
- Methodological: A novel self-supervised graph condensation framework that constructs latent pseudo-labels without ground-truth supervision, enabling robust condensation in label-scarce or noisy environments.

- Theoretical: Proofs establishing the stability of pseudo-label inference and bounds on embedding divergence. The authors show that under mild assumptions (sub-Gaussian latent structure and separability), the learned pseudo-labels concentrate around true latent centers and preserve cluster separation.

- Empirical: Extensive experiments on five benchmark datasets (Cora, Citeseer, Ogbn-arxiv, Flickr, Reddit) demonstrating that PLGC matches the performance of state-of-the-art supervised condensation methods on clean data and significantly outperforms them under high label noise, few-shot settings, and multi-source heterogeneous scenarios.

Key strengths:
- Well-motivated label-noise robustness angle with clear empirical evidence that supervised condensation methods degrade sharply under noise.
- Concentration-style theoretical analysis, which is uncommon in the graph condensation literature.
- Broad task coverage: clean, few-shot, multi-source, noise-robust, and link-prediction transfer.

Key weaknesses:
- No ablations on K, Sinkhorn ϵ, augmentations, or the swapped-assignment loss.
- Key self-supervised condensation baselines are missing or under-compared (CTGC only in 3-shot; ST-GCOND and SGDC not at all).

**Audience:**

Yes

**Audience Explanation:**

Graph Neural Networks (GNNs) and dataset condensation are highly active research areas within the TMLR community. As graph datasets grow to millions of nodes and edges, the computational cost of training becomes a significant bottleneck. This paper addresses a critical practical limitation of current state-of-the-art condensation methods—their fragility in the face of noisy or missing labels. The bridge between self-supervised learning and data compression is likely to interest researchers in graph ML, data efficiency, and robust representation learning.

**Broader Impact Concerns:**

None. The paper focuses on data efficiency and algorithmic robustness.

**Claims And Evidence:**

Yes

**Claims Explanation:**

The authors provide a comprehensive evaluation using five diverse benchmark datasets, ranging from small citation networks (Cora) to large-scale graphs (Reddit). They compare PLGC against a wide array of baselines, including supervised condensation (GCond, SFGC, GEOM), coarsening, and existing self-supervised methods (CTGC).

**Requested Changes:**

- [Critical] Provide a more detailed sensitivity analysis for key hyperparameters
- [Strengthening] Computational Efficiency Analysis (Strengthen): While the paper claims efficiency benefits over trajectory-based methods, it would be beneficial to include a direct comparison of training time and memory usage against other efficient baselines like DosCond or ST-GCond.
- [Strengthening] Heterogeneous Graphs: Briefly discuss how the framework might handle heterogeneous graphs (multiple node/edge types), as this was mentioned as a future direction but is a common real-world requirement.

---

### Decision · Action_Editor_ZAQr · 2026-06-05

**Recommendation:** Accept as is

**Audience:**

Yes

**Audience Explanation:**

Yes, researchers working on graph neural networks, graph condensation, self-supervised learning, and data-efficient training would likely find the paper relevant.

**Claims And Evidence:**

Yes

**Claims Explanation:**

Yes. The main claims are supported by theoretical analysis and empirical results across multiple graph benchmarks, including evidence for robustness under label noise and competitive performance against supervised condensation baselines. While some assumptions and comparisons could still be clearer, the revision appears to have addressed the major concerns through added ablations, efficiency analysis, and discussion of limitations.